# Synergistic and Regulatable Bioremediation Capsules Fabrication Based on Vapor-Phased Encapsulation of *Bacillus* Bacteria and its Regulator by Poly-*p*-Xylylene

**DOI:** 10.3390/polym13010041

**Published:** 2020-12-24

**Authors:** Yen-Ching Yang, Wei-Shen Huang, Shu-Man Hu, Chao-Wei Huang, Chih-Hao Chiu, Hsien-Yeh Chen

**Affiliations:** 1Department of Chemical Engineering, National Taiwan University, Taipei 10617, Taiwan; youngeddie.tw@gmail.com (Y.-C.Y.); sofiahu870628@gmail.com (S.-M.H.); jordanmagic0712@gmail.com (C.-W.H.); 2Institute of Oceanography, National Taiwan University, Taipei 10617, Taiwan; lowridertrent@gmail.com; 3Department of Orthopedic Surgery, Chang Gung Memorial Hospital, Taoyuan 33378, Taiwan; 4Bone and Joint Research Center, Chang Gung Memorial Hospital, Linkou 33305, Taiwan; 5Molecular Imaging Center, National Taiwan University, Taipei 10617, Taiwan; 6Advanced Research Center for Green Materials Science and Technology, National Taiwan University, Taipei 10617, Taiwan

**Keywords:** parylene, CVD, bioremediation, polymer, cellulase, *Bacillus*

## Abstract

A regulatable bioremediation capsule material was synthesized with isolated single-strain bacteria (*Bacillus* species, *B. CMC1*) and a regulator molecule (carboxymethyl cellulose, CMC) by a vapor-phased encapsulation method with simple steps of water sublimation and poly-*p*-xylylene deposition in chemical vapor deposition (CVD) process. Mechanically, the capsule construct exhibited a controllable shape and dimensions, and was composed of highly biocompatible poly-*p*-xylylene as the matrix with homogeneously distributed bacteria and CMC molecules. Versatility of the encapsulation of the molecules at the desired concentrations was achieved in the vapor-phased sublimation and deposition fabrication process. The discovery of the fabricated capsule revealed that viable living *B. CMC1* inhabited the capsule, and the capsule enhanced bacterial growth due to the materials and process used. Biologically, the encapsulated *B. CMC1* demonstrated viable and functional enzyme activity for cellulase activation, and such activity was regulatable and proportional to the concentration of the decorated CMC molecules in the same capsule construct. Impressively, 13% of cellulase activity increase was realized by encapsulation of *B. CMC1* by poly-*p*-xylylene, and a further 34% of cellulase activity increase was achieved by encapsulation of additional 2.5% CMC. Accordingly, this synergistic effectiveness of the capsule constructs was established by combining enzymatic *B. CMC1* bacteria and its regulatory CMC by poly-*p*-xylylene encapsulation process. This reported encapsulation process exhibited other advantages, including the use of simple steps and a dry and clean process free of harmful chemicals; most importantly, the process is scalable for mass production. The present study represents a novel method to fabricate bacteria-encapsulated capsule for cellulose degradation in bioremediation that can be used in various applications, such as wastewater treatment and transforming of cellulose into glucose for biofuel production. Moreover, the concept of this vapor-phased encapsulation technology can be correspondingly used to encapsulate multiple bacteria and regulators to enhance the specific enzyme functions for degradation of various organic matters.

## 1. Introduction

Bioremediation provides promising solutions for the removal of environmental pollutants, toxic elements, and poisoning management for clinical purposes [1,2,3]. Various remedial approaches were developed by precision screening of efficient bacteria from contaminated environments [4,5] to isolate genetically engineered bacteria with enhancement of catabolic pathways and microbial physiology [6,7,8]. To protect bacteria/cells from acidic environments and to elongate the treatment length, circulation time, and shelf life, different techniques, including spray drying/coating, freeze drying, emulsification, coacervation, extrusion, and microfluidics, have been developed for probiotic bacterial encapsulation [9,10]. However, the issue of the low vitality of the encapsulated bacteria used for bioremediation remains [11]. Imperatively, the technique used to create a microenvironment that can provide solid protection for bacteria, whose growth may be suppressed by parasites or predators [12] as well as control the release of either enzymes and/or bacteria to target pollution [13], is very important in the encapsulation of bacteria for bioremediation use. Therefore, the development of prospective encapsulation techniques for bioremediation has been pursued to provide additional properties that include (1) a favorable habitation microenvironment for functional bacteria with enhanced growth activities and (2) a regulatory mechanism to customize the enzymatic functions of bacteria, allowing increased efficiency and adjustment to the required bioremediation conditions.

We herein provide a vapor-phased encapsulation method for bacterial capsules with an elegantly equipped mechanism for increasing bacterial populations and the corresponding enzymatic functions. A vapor-deposited polymeric matrix system of poly-*p*-xylylenes of United States Pharmacopeia (USP) class VI with high biocompatibility and chemical resistance to strong acids, bases, and solvents was used for encapsulation during the fabrication process. The fabrication is performed in one step with a dry and clean vapor phase, which is desirable for sensitive biological substances such as cells, enzymes, growth factors, and other functional peptides and proteins [14,15]. The overall encapsulation process was realized based on our previously reported mechanism to deposit a vapor-phased poly-*p*-xylylene polymer on a template substrate that is eventually sublimated, and by manipulating the mass transport during the processing conditions, diverse species with distinct thermodynamic properties were subjected to sublimation and deposition within the confined space of the templates. Finally, transformation of the template resulted in the construction of composite materials with defined physical properties in terms of porosity, bulk size and geometry, and chemical functionality by the compartmentalized functional entities and the devised interfacial chemistries [16]. With respect to the vapor sublimation and deposition process, which has been reported to result in the precise localization and distribution of substances such as metals, molecules, and liquids to form composites with controlled homogeneity or anisotropy from various materials [15,17], we hypothesized that the benign and versatile vapor-phased process was able to encapsulate (i) unmodified native bacteria of the *Bacillus* species CMC1 (hereafter referred to as *B. CMC1*) with specific enzymatic function and the same precision to localize and distribute *B. CMC1* due to the encapsulation technique, enabling well-controlled bacterial growth activities and functions, and (ii) a regulator molecule, carboxymethyl cellulose (CMC), with a customizable encapsulation concentration to provide the same customizable stimulation dosages to regulate the enzymatic functions of the neighboring *B. CMC1* bacteria. The fabricated and encapsulated capsules were composed of inhabitant bacteria from (i) and the surrounding conditioning regulator molecules from (ii), and the synergistic activities exhibited by (i) and (ii) were able to deliver a combination of controlled enzymatic factors for bioremediation in the devised microenvironments (Figure 1a). The synergistic conditioning mechanism was effective in regulating the bacterial number and the corresponding enzymatic reactivity, and was proportional to the CMC regulator composition. The encapsulation process is versatile and is completed in simple steps with mass production scalability (Figure 1a,e), and the resultant bacteria/CMC capsules represent an advanced bacteria/polymer combination for bioremediation models and have potential for unlimited combinatorial composting configurations for bioremediation applications.

## 2. Materials and Methods

### 2.1. Bacterial Strain Isolation

Wastewater and activated sludge were collected from the sewage of a pig farm in Taoyuan, Taiwan. All tools and reservoirs were sterilized before sampling. Activated sludge was stored immediately at 4 °C after collection. The wastewater and activated sludge were used for further culturing of environmental bacteria in liquid nutrient medium. In one liter of culture medium, 3 g peptone (BD biosciences, San Jose, CA, USA), 1 g yeast extract (BD biosciences, USA), 5 g NaCl (Sigma-Aldrich, St. Louis, MO, USA), 0.2 g MgSO_4_·H_2_O (Sigma-Aldrich, USA), 0.2 g CaCl_2_ (Sigma-Aldrich, USA), 0.2 g KNO_2_ (Sigma-Aldrich, USA), 2 g NH_4_NO_3_ (Sigma-Aldrich, USA), 0.3 g Na_2_HPO_4_ (Sigma-Aldrich, USA), 5 g glucose (Sigma-Aldrich, USA), and 2.5 mL tween 80 (Sigma-Aldrich, USA) were added and then pH with NaOH to 7.0. The genomic DNA of the isolated bacterial strain was extracted using a commercial genomic DNA extraction kit (Qiagen, Germantown, MD, USA). Then, the sequence of its 16S rRNA was confirmed by polymerase chain reaction by using the primers F8 (5′GAGAGTTTGATCCTGGCTCAG3′) and R1492 (5′GGTTACCTTGTTACGACTT3′). The obtained 16S rRNA sequence from the isolated bacteria was

TGCTATACATGCAGTCGAGCGGACAGATGGGAGCTTGCTCCCTGATGTTAGCGGCGGACGGGTGAGTAACACGTGGGTAACCTGCCTGTAAGACTGGGATAACTCCGGGAAACCGGGGCTAATACCGGATGGTTGTTTGAACCGCATGGTTCAAACATAAAAGGTGGCTTCGGCTACCACTTACAGATGGACCCGCGGCGCATTAGCTAGTTGGTGAGGTAACGGCTCACCAAGGCAACGATGCGTAGCCGACCTGAGAGGGTGATCGGCCACACTGGGACTGAGACACGGCCCAGACTCCTACGGGAGGCAGCAGTAGGGAATCTTCCGCAATGGACGAAAGTCTGACGGAGCAACGCCGCGTGAGTGATGAAGGTTTTCGGATCGTAAAGCTCTGTTGTTAGGGAAGAACAAGTACCGTTCGAATAGGGCGGTACCTTGACGGTACCTAACCAGAAAGCCACGGCTAACTACGTGCCAGCAGCCGCGGTAATACGTAGGTGGCAAGCGTTGTCCGGAATTATTGGGCGTAAAGGGCTCGCAGGCGGTTTCTTAAGTCTGATGTGAAAGCCCCCGGCTCAACCGGGGAGGGTCATTGGAAACTGGGGAACTTGAGTGCAGAAGAGGAGAGTGGAATTCCACGTGTAGCGGTGAAATGCGTAGAGATGTGGAGGAACACCAGTGGCGAAGGCGACTCTCTGGTCTGTAACTGACGCTGAGGAGCGAAAGCGTGGGGAGCGAACAGGATTAGATACCCTGGTAGTCCACGCCGTAAACGATGAGTGCTAAGTGTTAGGGGGTTTCCGCCCCTTAGTGCTGCAGCTAACGCATTAAGCACTCCGCCTGGGGAGTACGGTCGCAAGACTGAAACTCAAAGGAATTGACGGGGGCCCGCACAAGCGGTGGAGCATGTGGTTTAATTCGAAGCAACGCGAAGAACCTTACCAGGTCTTGACATCCTCTGACAATCCTAGAGATAGGACGTCCCCTTCGGGGGCAGAGTGACAGGTGGTGCATGGTTGTCGTCAGCTCGTGTCGTGAGATGTTGGGTTAAGTCCCGCAACGAGCGCAACCCTTGATCTTAGTTGCCAGCATTCAGTTGGGCACTCTAAGGTGACTGCCGGTGACAAACCGGAGGAAGGTGGGGATGACGTCAAATCATCATGCCCCTTATGACCTGGGCTACACACGTGCTACAATGGACAGAACAAAGGGCAGCGAAACCGCGAGGTTAAGCCAATCCCACAAATCTGTTCTCAGTTCGGATCGCAGTCTGCAACTCGACTGCGTGAAGCTGGAATCGCTAGTAATCGCGGATCAGCATGCCGCGGTGAATACGTTCCCGGGCCTTGTACACACCGCCCGTCACACCACGAGAGTTTGTAACACCCGAAGTCGGTGAGGTAACCTTTTAGGAGCCAGC.

The sequence was further confirmed according to the 16S rRNA database at https://www.ezbiocloud.net/. The isolated bacteria were the *Bacillus* species with the top ranking of *B. subtilis* and was named *Bacillus* species CMC1 (*B. CMC1*) in the study.

### 2.2. Encapsulation Process

*B. CMC1* was cultured for 30 h at 30 °C and then washed twice with diH_2_O by centrifugation and resuspended into final solutions (either diH_2_O, 0.25% CMC (Showa, Tokyo, Japan), 1.5% CMC, or 2.5% CMC) to prepare samples with the same concentration of bacteria (~1 × 10^8^ bacteria/mL) on the same day of capsule fabrication. The prepared bacterial solution with/without CMC was solidified by liquid nitrogen for later encapsulation. A home-built vapor deposition system was used for the encapsulation and fabrication process in this study (illustrated in Figure 1e), and the specifications and detailed installations of the system are described elsewhere [18,19]. First, the precursor dichloro-[2,2]-paracyclophane (Galxyl C, Galentis, Venice, Italy) was transformed into di-radical monomers by pyrolysis at 670 °C for the subsequent vapor deposition of poly-*p*-xylylenes. Subsequently, the monomers started both polymerization and deposition under the conditions of 100 mTorr and 25 °C. Simultaneously, the solid ice in the bacterial ice templates was transferred into water vapor via a sublimation process with respect to the same conditions of 100 mTorr and 25 °C. The deposition of poly-*p*-xylylenes and sublimation of solid ice into water vapor produced porous poly-*p*-xylylene bacteria capsules of the desired size and shape.

### 2.3. Enzyme Activity Assays

Cellulase activity was examined on 9 cm agar plates by quantifying the size of the enzyme functioning zones. To one liter of cellulase visualizing agar, 1 g yeast extract (BD Biosciences, USA), 1 g NH_4_H_2_PO_4_ (Sigma-Aldrich, USA), 0.2 g KCl (Sigma-Aldrich, USA), and 1 g MgSO_4_·7H_2_O (Sigma-Aldrich, USA) were added, and then the pH was adjusted to 6.3 with NaOH. Then, 9 g agar (Sigma-Aldrich, USA) and 10 mg trypan blue (Sigma-Aldrich, USA) were added. A single bacterial capsule (100 µL in volume) was seeded in the center of one 9 cm plate and left at 30 °C for up to 140 h for subsequent enzyme functioning zone quantification. For the quantification of DNA and RNA, a single bacterial capsule (100 µL in volume) was seeded in a 50 mL centrifuge tube with 30 mL of growth culture medium inside, and the culture was left at 30 °C for 50 h on an orbital shaker with a 100 rpm shaking speed. The same volume of the culture (5 mL) from encapsulated bacteria was centrifuged for genomic DNA extraction using a commercial extraction kit (Zymo Research, Irvine, CA, USA) for the following DNA quantification. The same number of bacterial cells (2 × 10^8^ cells) was used for total RNA extraction by using a commercial extraction kit (Zymo Research, USA) for total RNA quantification. The concentrations of DNA and RNA were measured by a NanoDrop Spectrophotometer (ThermoFisher, Waltham, MA, USA). For this, 1 µL of purified DNA and RNA were used in the analysis. Note that only the DNA and RNA with the 260/280 ratio between 1.8 and 2.0 were used for concentration measurements, ensuring the sufficient purity of DNA and RNA for further quantification analysis.

### 2.4. Characterizations

A Nikon ECLIPSE 80i fluorescence microscope (Nikon, Tokyo, Japan) with a visible light source was used to visualize the crystal violet-stained *B. CMC1* in the porous parylene structure. A VK-9510 3D profile microscope (Keyence, Osaka, Japan) was used to analyze the external architecture of the bacterial capsules and the existence of the bacteria inside the structure. SEM images were recorded using a NovaTM NanoSEM (FEI, Hillsboro, OR, USA) operated at a primary energy of 10 keV and a pressure of 5 × 10^−6^ Torr to detailed the internal structure of the bacterial capsule. EDS elemental point analyses were captured for the quantification of the studied elements. 3D analysis of the interior structure was performed by using Bruker Skyscan 2211 (Bruker micro-CT, Kontich, Belgium) at 2.0 μm/pixel resolution. The setting of the voltage was 40 kVp, whereas the current was 700 μA at 8 Watt output with microfocus mode. Image reconstruction, ring artifacts, and beam-hardening correction were performed using reconstruction software, Instarecon (Bruker micro-CT, Belgium). FT-IR spectra of the fabricated capsules were recorded by a Spectrum 100 spectrometer equipped with an ATR detector (PerkinElmer, Waltham, MA, USA). The recorded spectra ranged from 600 to 4500 cm^−1^ with 4 scan times at 4 cm^−1^ resolution.

## 3. Results and Discussion

### 3.1. The Fabrication of Regulatable Bioremediation Capsules

The bacteria-encapsulated capsule was fabricated by first preparing ice templates by directly transforming the liquid-phased bacteria-cultured media into a solidified ice template in a liquid nitrogen-conditioned bath. Subsequently, the fabrication exploited the previously described sublimation of ice and deposition of poly-*p*-xylylene in one continuous step [16,17] to construct a bacterially encapsulated polymeric capsule. A second transformation was performed with vapor-deposited poly-*p*-xylylene molecules to replace the resulting space when the sublimated ice/water molecules evaporated from the ice templates. The resultant construct consisted of a three-dimensional porous poly-*p*-xylylene matrix with encapsulated *B. CMC1* bacteria in the matrix (Figure 1). The shape of the final capsule construct replicated the shape of the transformed ice template, which was theoretically shaped by molding, sculpting, or solidifying droplets to obtain various sizes and geometries [15]. Ideally, shrinkage or dislocation of the construct is avoided owing to the continuous sublimation and deposition process [16]. As a result, the dimension of the fabricated capsule is depended on the size and shape of the mold used in the ice template preparation. The present technology supports to fabricate the bacteria-encapsulated capsule as little as 400 µm cubic shape to a larger 5mm columnal shape. As shown in Figure 1b, a cubic shape with dimensions of 400 µm × 400 µm × 400 µm was used for the preparation of the ice templates and the fabrication of the construct of the bacterial-encapsulated capsule. The subsequent CVD process replaced ice with poly-*p*-xylylene to form a three-dimensional porous structure (Figure 1c). Characterization by microcomputed tomography (micro-CT, Figure 1d) also indicated the overall homogeneous porous structure of the polymer matrix. By using the column-shaped mold with the dimension of 5 mm in diameter and 5 mm in height, the larger size of bacteria-encapsulated capsule was obtained (Figure 1f).

In order to characterize the internal structure of the bacteria encapsulated capsule, the 400 µm cubic capsule was examined by using a combination of optical microscopy, scanning electron microscopy (SEM), and confocal microscopy to verify the localization and distribution of the bacteria (Figure 2). The image recorded by optical microscopy clearly showed that the crystal violet-stained bacteria were found only in the polymer matrix but not in the void pores, verifying the successful encapsulation of bacteria by the fabrication process (Figure 2a). Both confocal microscopy and SEM further confirmed that rod-shaped *B. CMC1* with dimensions of 0.5–1 µm existed in the fabricated capsule structures (Figure 2a). The anticipated homogeneity of the distributed bacteria was also observed and was believed to be due to the controlled mass transport of solidified ice templates and the continued sublimation and deposition process that prevented dislocation of the encapsulated substances. As the same concentration of bacteria was used to prepare the ice template, the quantity of the bacteria was depended on the volume of the ice template prepared for the capsule fabrication. Therefore, the larger column-shaped bacteria-encapsulated capsules which contained sufficient number of bacteria in the fabricated capsules were used for cellulase activities examination on 9 cm agar plates.

### 3.2. Viability and Cellulase Activities of Encapsulated B. CMC1

The cell viability of the encapsulated bacteria was further analyzed, and due to the natural self-protective mechanism of the cell wall in *B. CMC1*, we hypothesized that the bacteria were resistant to low temperatures during iced template preparation and low-pressure conditions (approximately 10^−3^ Torr) during the sublimation and deposition process. Cultured samples of encapsulated *B. CMC1* were compared to samples of *B. CMC1* without encapsulation (positive control), and their cell densities were monitored by OD_600_ measurement. The results showed comparable bacterial growth patterns and activities in liquid medium for both types of bacteria during the culture time frame of 10 h (Figure 2b), indicating the viability of cells and negligible disturbance of *B. CMC1* by the encapsulation process. In addition, by comparing the enzymatic function for cellulase decomposition on 9.0 cm agar plates for both groups of samples, the data at 140 h showed a 4.3 cm enzyme functional zone diameter for the encapsulated *B. CMC1* compared to a 3.8 cm diameter for samples without encapsulation (Figure 2c). Unambiguously, these results were found to support the hypothesis that the *B. CMC1* bacteria were preserved during the encapsulation and fabrication process. Interestingly, it was discovered that a higher population and a larger enzyme effective zone size were found for the *B. CMC1*-encapsulated samples, which indicated their enhanced bacterial growth and cellulase enzymatic activities compared to those of the nonencapsulated sample at the studied time point of 140 h, and it is believed that the highly biocompatible poly-*p*-xylylene [20,21,22] played an important role and provided an agreeable microenvironmental niche for *B. CMC1*.

### 3.3. Regulation of the Fabricated Capsules by CMC

CMC was found to enhance the cellulase activity of *Bacillus* species bacteria [23]. Here, in this study, the proposed regulation approach exploited the cellulase self-induction mechanism by supplying a various number of CMC molecules for *B. CMC1* in the same controlled volume in the capsule construct, and our second hypothesis was thus raised to validate whether increasing the concentration of CMC would proportionally increase the resultant enzymatic activity of *B. CMC1* in the same capsule construct. While the versatility of the sublimation and deposition process was able to encapsulate/include multiple ingredients with a precisely administered composition and without molecular phase separation [16], the fabrication was straightforward and conducted by preparing iced templates containing both (i) *B. CMC1* and (ii) various concentrations of CMC (0.25%, 1.5%, or 2.5% were selected for the demonstration in the study). Subsequently, the same sublimation and deposition process was performed to resolve the capsule construct that comprised a poly-*p*-xylylene matrix with simultaneous encapsulation (i) and (ii) the same matrix structure. In addition to the already confirmed encapsulation of *B. CMC1*, the use of FT-IR analysis indicated that the intensity of the characteristic –C=O and –CH–O–CH_2_ peaks from CMC observed at 1558 and 1050 cm^−1^, respectively, gradually increased from a low concentration to higher concentrations of CMC content in the fabricated capsules (Figure 3a). Moreover, quantification of the peak area for –C=O and –CH–O–CH_2_, which were normalized with respect to nonshifted peaks (–C–H at approximately 2800 cm^−1^) from poly-*p*-xylylene, additionally showed consistency by stoichiometry of the encapsulated CMC concentrations (Figure 3b). On the other hand, the consistent increases in elemental concentrations of oxygen (0% to 27.98%) and sodium (0% to 4.73%) and decreases in carbon (95.22% to 65.25%) and chloride (4.78% to 2.03%) in SEM-EDS analysis were also in close value to the theory, which also verified the proposed encapsulation and regulation of CMC compositions (Figure 3c). The SEM and optical microscopic images of bioremediation capsules containing various concentrations of CMC (0.25%, 1.5%, 2.5%) did not show any obvious difference in the porous structure within the capsules. This is because the poly-*p*-xylylene matrix replaced sublimated water to constitute the main structure of the capsule during the vapor encapsulation process, the poly-*p*-xylylene matrix might deposited directly on the CMC molecules when water sublimated. The function of CMC molecular in these bioremediation capsules was only to upregulate the function of cellulase activity [23]. However, the pore size of the capsule can be controlled by the deposition rate of the poly-*p*-xylylene [16].

With respect to the effectiveness of increasing CMC compositions toward the growth activities of *B. CMC1* in the same fabricated capsules, separate experiments were performed by culturing *B. CMC1* in liquid medium for samples with varied CMC compositions and were compared to the samples without CMC encapsulation. As shown in Figure 4a, the cultured regulation capsule samples (both *B. CMC1* and CMC were encapsulated) exhibited approximately 10 h of bacterial growth in the lag phase and showed consistency with the samples with only *B. CMC1* encapsulation (Figure 2b), indicative of negligible disturbance by the surrounded and encapsulated CMC molecules, and the bacteria adapted themselves well to the engineered microenvironments of the capsules. Interestingly, and surprisingly, the encapsulation of CMC molecules showed a positive influence on the growth of *B. CMC1* in the exponential phase; the presence of CMC resulted in the enhancement of *B. CMC1* growth activities, and such an enhancement was found to be proportional to the increase in CMC composition. Two stages in the exponential phase of showing a steep growth curve in the early exponential growth phase from approximately 10–15 h and a second gradual incline stage from 15–40 h were found. The bacterial growth finally reached a stationary phase after 40 h, and due to the regulation and enhancement of varied CMC compositions, increasing the final bacterial number was found accordingly for the studied groups.

The important expressions of regulated enzyme activities were finally analyzed for the synergistic capsules. Capsule samples containing both *B. CMC1* and CMC (with the same varied compositions of 0.25%, 1.5%, and 2.5%) were studied in parallel to bare samples with unregulated *B. CMC1* (0% CMC), and their enzymatic cellulase activities were first examined on agar plate results for up to 140 h (enzyme functioning zone measuring collected from three independent experiments were listed in Table 1). As shown in Figure 4b,c, comparisons of the size of the enzyme functioning zones were measured and quantified for these plates, and anticipated results indicated that increased zone sizes were found with increasing CMC compositions. More specifically, the quantification of the zone size revealed a 9.7% increase from 0% to 0.25% of CMC composition variation and a 26.5% and 35.6% increase for 1.5% and 2.5%, respectively, demonstrating a regulatable and enhanced enzymatic cellulase activity of the synergistic capsules in accordance with the encapsulated variation of CMC compositions.

More evidence showing the enhancement of the enzyme activities was further verified through the analysis of the genomic information, including the amount of genomic DNA and total RNA from the cultured samples of the capsules at a time point of 50 h, which was in the early stage of stationary phase in the bacterial growth curve. Calculation and quantification of the genomic DNA was performed based on the same volume of bacteria obtained from the capsule samples, and as shown in Figure 4d, an increase by 7.5% of the DNA concentration from unregulated *B. CMC1* (0% CMC) to a 0.25% regulation was found, and an anticipated enhancement of the expression with increasing CMC regulation compositions was accordingly discovered with a 13.1% increase for 1.5% CMC and 27.4% for 2.5% CMC of the studied capsule samples. On the other hand, based on the same bacteria number to extract their total RNA, the results in Figure 4e also indicated a similar trend with anticipation, with a 9% increase for 0.25% CMC, 14% increase for the 1.5% CMC sample, and 18.4% increase for the 2.5% CMC capsules. The genomic information specifically verified the enzymatic ability of *B. CMC1* was regulated for the synergistic capsules, and the synergy was found to be consistent with the same regulatable and enhancement of the bacterial population in the aforementioned study. Collectively, the results unambiguously verified the hypothesis that synergic effectiveness was achieved by vapor encapsulation of (i) *B. CMC1* and (ii) the CMC regulator in the same fabricated capsule constructs, and the regulation was achievable through the versatile use of the encapsulated CMC compositions.

## 4. Conclusions

Synergistic and regulatable bioremediation of cellulose by encapsulated *B. CMC1* was achieved in the current study. Because of the versatility of various types of remedial bacteria and regulator molecules, unlimited applications are expected beyond those shown in the report. In addition, the use of water/ice templates and a dry and clean vapor-phased process preserved the sensitive biomolecules and their biological functions, and the final fabricated capsule construct was composed of a USP Class VI compatible poly-*p*-xylylene matrix. Due to this specific production process, this novel fabrication process can not only encapsulate functional bacteria, regulating molecules, but also other potential absorbing material to concentrate the organic matters for degradation. It was reported that the encapsulation of phenol metabolic bacteria in microfiltration membrane capsules alone can create a confined environment to enhance the bioremediation of efficiency of encapsulated bacteria [24]. The poly-*p*-xylylene fabrication technology can constitute a defined inner porous structure to capture different size of organic particles to further concentrate organic matters and a controllable size and shape of the products to fit to different situations. Until now, most of the present studies still focused on identifying the bacteria with specific enzyme functions to use in the bioremediation process [5,25,26]. We foresee the application of capsule products to the degradation of organic compounds, wastewater, and environmental pollution, as well as the removal of potential hazardous chemicals.

## Figures and Tables

**Figure 1 polymers-13-00041-f001:**
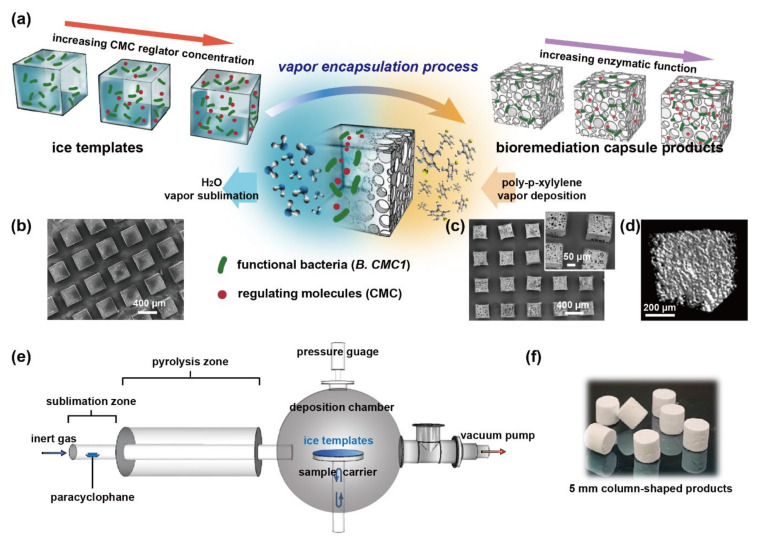
Fabrication of regulatable bioremediation capsules via vapor-phased encapsulation of bacteria. (**a**) Schematic illustration of the capsule fabrication process by vapor sublimation and deposition to encapsule (i) functional bacteria and (ii) regulatory molecules within a poly-*p*-xylylene polymer matrix. (**b**) Image by cryo SEM showing the iced templates of array cubes with dimensions of 400 µm × 400 µm × 400 µm used for encapsulation. (**c**) SEM images of the capsule products by the encapsulation process; the products showed the same replicated dimension and shape and comprised poly-*p*-xylylene as the matrix and encapsulated bacteria and CMC molecules. Panels (**b**,**c**) show the scalability of mass production potentials. (**d**) A 3-D image by micro-CT examination of the capsule product. (**e**) An illustration of the vapor deposition system, containing a sublimation zone, a pyrolysis zone, and a deposition chamber in the main body. (**f**) The fabricated column-shaped bioremediation capsule products with the dimension of 5 mm in diameter and 5 mm in height.

**Figure 2 polymers-13-00041-f002:**
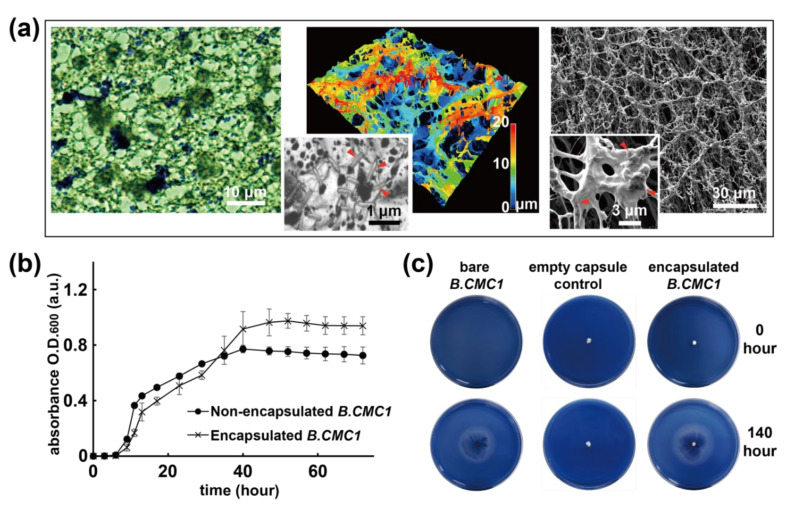
Viability analysis of *B. CMC1* activities of the fabricated capsule. (**a**) Micrographs showing stained (crystal violet, purple color), 3-D structural, and SEM topological images of encapsulated *B. CMC1* within the capsule products. The yellow marks indicate the presence of *B. CMC1* within the polymer structure. (**b**) Bacterial growth curves and (**c**) enzymatic functions were compared in parallel for pure bacteria to verify viability.

**Figure 3 polymers-13-00041-f003:**
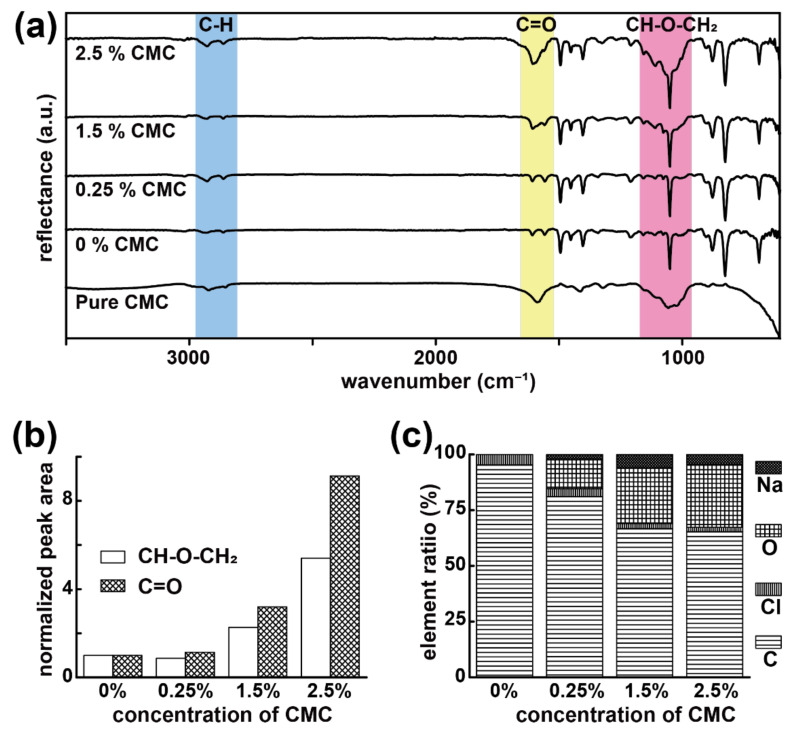
Regulation and encapsulation of CMC molecules. (**a**) FT-IR analysis showed regulated concentrations of CMC in the capsule products. (**b**) Quantified and normalized peak intensity of C=O and CH-O-CH_2_ in panel (**a**) indicated a consistent increase according to the increased stoichiometric concentrations of CMC. (**c**) EDS elemental analysis revealed similar trends in the carbon and oxygen composition variations corresponding to the regulations of CMC concentrations.

**Figure 4 polymers-13-00041-f004:**
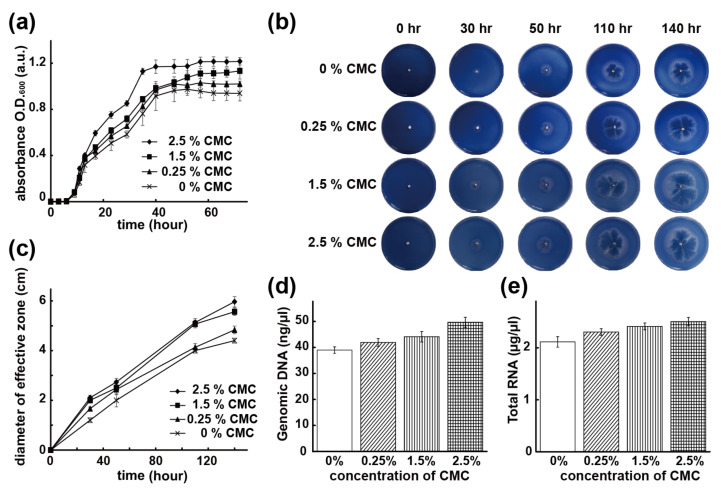
Regulation and quantification of bacterial biochemical activities of the bacterial capsules. (**a**) Regulation to enhance bacterial growth with respect to increasing CMC concentrations was verified in a studied culture time frame. (**b**) Regulated and enhanced enzyme activities of *B. CMC1* were discovered with increasing CMC concentrations on agar plates for up to 140 h. (**c**) Quantification of the effective zones in panel (**b**) was compared. (**d**) Analysis of the genomic DNA for the studied groups confirmed the upregulated expression of the corresponding DNA according to the increments of CMC concentrations. (**e**) The same upregulated expression of total RNA unambiguously verified the proposed regulation mechanism by increasing CMC concentrations in the bacterial capsules. Statistical data were expressed as the mean value and the standard deviation based on three independent samples.

**Table 1 polymers-13-00041-t001:** The diameter of enzyme functioning zones recorded in the enzymatic cellulase examination in Figure 4b,c.

**0% CMC**	**Time (h)**	**Diameter of the Functioning Zone (cm)**			**0.25% CMC**	**Time (h)**	**Diameter of the Functioning Zone (cm)**		
		**Exp. 1**	**Exp. 2**	**Exp. 3**			**Exp. 1**	**Exp. 2**	**Exp. 3**
	0	0	0	0		0	0	0	0
	30	1.2	1.1	1.3		30	1.6	1.7	1.7
	50	1.9	1.8	2.3		50	2.6	2.2	2.5
	110	3.9	4.1	4		110	4	4.1	4.3
	140	4.3	4.5	4.4		140	5	4.8	4.7
**1.5% CMC**	**Time (h)**	**Diameter of the Functioning Zone (cm)**			**2.5% CMC**	**Time (h)**	**Diameter of the Functioning Zone (cm)**		
		**Exp. 1**	**Exp. 2**	**Exp. 3**			**Exp. 1**	**Exp. 2**	**Exp. 3**
	0	0	0	0		0	0	0	0
	30	2	2.1	1.9		30	2	2.2	2.1
	50	2.5	2.4	2.6		50	2.7	2.9	2.6
	110	5.1	4.9	5.2		110	5.1	5.3	5
	140	5.6	5.4	5.7		140	5.9	6.2	5.8

## Data Availability

Data available on request due to privacy.

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
