# Peer review of "Synergistic and Regulatable Bioremediation Capsules Fabrication Based on Vapor-Phased Encapsulation of Bacillus Bacteria and its Regulator by Poly-p-Xylylene"

_polymers, 2020, doi:10.3390/polym13010041_

Round 1

Reviewer 1 Report

Dear authors

First of all, let me congratulate you for a so interesting and well developped work. Introduction gives enough information about the techniques and aims used and planned into the paper.

Results are enough to sustain your hypotheses and corroborate everything established previously.

In my opinion, the article is ready to be published and it will be many times referred due to its high level

Many thanks for your contribution

Author Response

Point 1: First of all, let me congratulate you for a so interesting and well developed work. Introduction gives enough information about the techniques and aims used and planned into the paper. Results are enough to sustain your hypotheses and corroborate everything established previously. In my opinion, the article is ready to be published and it will be many times referred due to its high level. Many thanks for your contribution

Response 1: We appreciate the comments of the reviewer and are very delighted with the positive feedback.

Reviewer 2 Report

The title of the manuscript has to be modified. It needs to clearly indicate the type of matrix within the obtained capsules as well as the type of bacteria used in the study.

(line 22) The Authors indicate that a „vapor-phased encapsulation method with simple steps” was used. The term “simple steps” has to be explained.

In the “Abstract” part of the work, the most important results have to be mentioned. A statement such as “The fabricated encapsulation capsule represents an advanced bioremediation material for multidisciplinary fields and applications.” (lines 35-36) is not sufficient.

(line 135) The Authors indicate that “A home-built vapor deposition system was used”. The schematic or a picture of the apparatus used in the study has to be presented. Moreover, a scheme of the encapsulation process should be described.

The Authors claim that “The concentrations of DNA and RNA were measured by a NanoDrop Spectrophotometer” (lines 159-160) More details of the measurement technique have to be presented in order to validate it.

The Authors indicate that “Cellulase activity was examined on 9 cm agar plates by quantifying the size of the enzyme functioning zones”. The images of the agar plates taken during the analysis of the cellulase activity should be added as supplementary materials.

The characteristics such as structure, shape, and dimensions of the obtained capsules are extremely important. For this reason taking into account that in an aim to analyze the obtained capsules, the following microscope methods such as: fluorescence microscope, profile microscope, SEM, and 3D analysis should be applied to all types of the obtained capsules. Moreover, the influence of the concentration of CMC on the mentioned parameters has to be described.

Considering that vital results have been omitted I suggest a major revision of the submitted manuscript.

Reviewer 3 Report

It is a very good study with overall adequate presentation of experimental results. Some additions are needed:

1) Authors should further emphasize on the novelty of their work.

2) Some minor typos, grammar and syntax errors should be carefully revised and corrected accordingly.

3) Reference can be even more updated (more recent relative works).

4) A major drawaback is the almost absence of References in Results and discussion. The paper has 22 Refs and from which 17 are in Intorduction. So, authors must further comment some already publshed works and enrich this part. 
